

# High temporal resolution wet delay gradients estimated from multi-GNSS and microwave radiometer observations

Tong Ning[1] and Gunnar Elgered[2]

[1]Lantmäteriet (The Swedish Mapping, Cadastral and Land Registration Authority), SE-80182, Gävle, Sweden
[2]Department of Space, Earth and Environment, Chalmers University of Technology, Onsala Space Observatory, SE-43992 Onsala, Sweden.

**Correspondence:** T. Ning (tong.ning@lm.se)

**Abstract.** We have used one year of multi-GNSS observations at the Onsala Space Observatory on the Swedish west coast to estimate the linear horizontal gradients in the wet propagation delay. The estimated gradients are compared to the corresponding ones from a microwave radiometer. We have investigated different temporal resolutions from 5 min to one day. Relative to the GPS-only solution and using an elevation cutoff angle of 10° and a temporal resolution of 5 min the improvement obtained for

the solution using GPS, Glonass, and Galileo data is an increase in the correlation coefficient of 11 % for the east gradient and 20 % for the north gradient. Out of all the different GNSS solutions, the highest correlation is obtained for the east gradients and a resolution of 2 h, while the best agreement for the north gradients is obtained for 6 h. The choice of temporal resolution is a compromise between getting a high correlation and the possibility to detect rapid changes in the gradient. Due to the differences in geometry of the observations, gradients which happen suddenly, are either not captured at all or captured but

with much less amplitude by the GNSS data. When a weak constraint is applied in the estimation of process, the GNSS data have an improved ability to track large gradients, however, at the cost of increased formal errors.

## 1 Introduction

An accurate modelling of the atmospheric effects on GNSS observations is relevant both for geodetic and meteorological applications, in forecasting as well as in climate research. In geodetic applications the standard method is to estimate an

equivalent zenith propagation delay together with a linear horizontal gradient. Early results showed an improved repeatability for the estimated coordinates when estimating gradients using GPS data (Bar-Sever et al., 1998; Meindl et al., 2004). A recent study (Zhou et al., 2017) found that estimating gradients with a temporal resolution of 1 h can achieve even better positioning performance than strategies where gradients are estimated with resolutions of many hours up to one day.

In meteorological applications the Zenith Total Delay (ZTD) and horizontal gradients may be assimilated directly into the

forecasting model, see e.g., (Zus et al., 2019). Inferred values of the Zenith Wet Delay (ZWD) and the Integrated Water Vapour (IWV) may be used to study long terms trends (Baldysz et al., 2018). Linear horizontal gradients estimated from GNSS have been used to study specific meteorological conditions. For example, on the island of Corsica, where the physical meaning of gradients in coastal areas with a steep topography was studied (Morel et al., 2015), and in Texas, USA, where significant gradients during the hurricane Harvey were reported (Graffigna et al., 2019).





The quality of the estimated gradients has been assessed by comparisons to independent measurements, such as using a microwave radiometer, in the following this instrument is referred to as a Water Vapour Radiometer (WVR), the space geodetic technique of very long baseline interferometry (VLBI), and numerical weather models.

Such an assessment was carried out by Elgered et al. (2019) where the GPS-derived gradients were compared with the ones obtained from WVR, VLBI, and the European Centre for Medium-Range Weather Forecasts (ECMWF) analyses. The results
show that the best agreement is obtained when an elevation cutoff angle of 3° is applied in the GPS data processing, in spite of the fact that the radiometer did not observe below 20°. They also found that a homogeneous and frequent sampling of the sky is a critical parameter for gradient estimation. Using multi-GNSS observations instead of GPS only Li et al. (2015) found a significant increase in the correlation coefficient from below 0.5 to about 0.6 when compared to the gradients computed from the ECMWF reanalysis product. The corresponding decrease in the root-mean-square (RMS) difference of the gradients was
25–35 % for multi-GNSS processing. The temporal resolutions of such comparisons are to our knowledge so far limited to 1 h for WVRs (Lu et al., 2016), 2 h for VLBI (Steigenberger et al., 2007), and 6 h for numerical weather models (Zus et al., 2019).

The aim of this study is to assess the quality of estimated gradients from multi-GNSS with temporal resolutions as high as 5 min using independent WVR data. Section 2 describes how the gradients are estimated from the GNSS and the WVR data. In Section 3.1 we present the gradients estimated for different GNSS constellations and different elevation cutoff angles.
These are thereafter compared to the WVR data. First in Section 3.2 for the highest temporal resolution of 5 min and then in Section 3.3, over time scales up to one day, both for the entire data set and for a specific event of short lived gradients associated with rapid changes in the ZWD. In Section 3.4 we also study the impact of using a weaker constraint for the random walk process of the gradient time series. Finally, Section 4 gives our conclusions.

## 2   Data sets

### 45  2.1  GNSS

We have analysed one year (1 Jan.–31 Dec., 2019) of ground-based GNSS observations acquired from one station (ONS1) located at the Onsala Space Observatory, on the west coast of Sweden. For comparison purposes we also used GNSS data from June and July 2019 acquired at the collocated station ONSA. The data processing was carried out using GipsyX v.1.5 (https://gipsy-oasis.jpl.nasa.gov/gipsy/docs/releaseNotes-GipsyX-1.5.pdf) with the Precise Point Positioning (PPP) strategy (Zum-
berge et al., 1997). The input to the processing was ionospheric free linear combinations formed by acquired GNSS phase-delay observations while the output included station coordinates, clock biases, and tropospheric parameters. The final multi-GNSS orbit and clock products used were provided by Center for Orbit Determination in Europe (CODE) (Prange et al., 2020). An ocean tide loading correction using the FES2004 model (Lyard et al., 2006) was applied while no atmospheric pressure loading corrections were used. The absolute calibration of the Phase Centre Variations (PCV) for all antennas (from the file
igs14_2101.atx) was implemented (Schmid et al., 2007). We used the Vienna Mapping Function 1 (VMF1) (Boehm et al., 2006) to map the zenith delay and the gradient mapping function was the one presented by Bar-Sever et al. (1998).





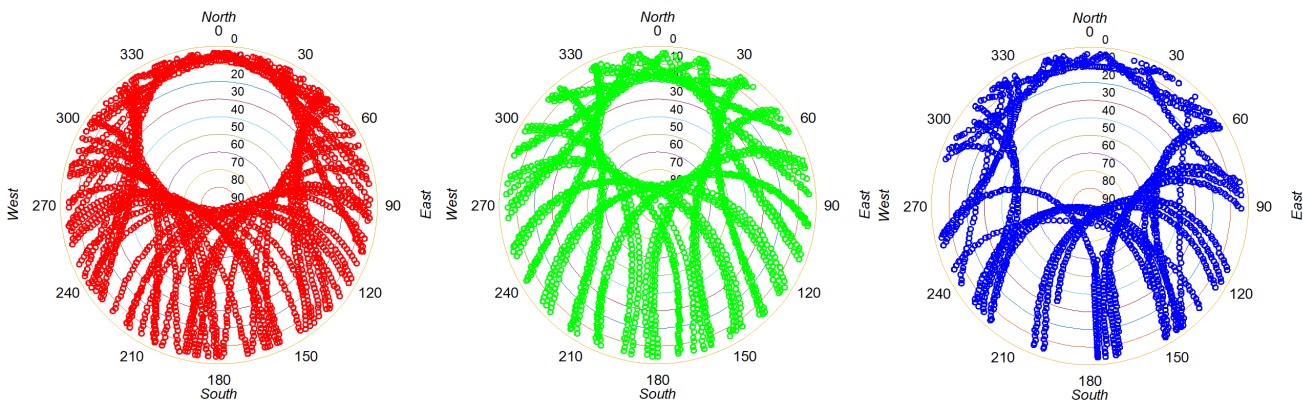

**Figure 1.** Observations on the sky acquired from ONS1 for 0 to 24 h on July 24, 2019, for GPS (left), Glonass (middle), and Galileo (right).

The ZTD and linear horizontal delay gradients were estimated every 5 min using a random walk model with a standard deviation (SD) of 10 mm$\sqrt{\text{h}}^{-1}$ and 0.3 mm$\sqrt{\text{h}}^{-1}$, respectively. No weighting was applied. The SD value used for the ZTD is given by Jarlemark et al. (1998) where they found a temporal variability in the wet delay, derived from 71 days of microwave

radiometer measurements, varying in the interval 3–22 mm$\sqrt{\text{h}}^{-1}$. Because our focus here is on high temporal resolution and that we expect wet gradients to sometimes be short lived we also use a looser constraint (in Section 3.4) of 1.0 mm$\sqrt{\text{h}}^{-1}$ for the SD in the random walk model for the gradients.

The Zenith Hydrostatic Delay (ZHD) was calculated using ground pressure measurements (Saastamoinen, 1973) and thereafter the ZWD was obtained by subtracting the ZHD from the ZTD. The gradients estimated from the GNSS data are total

gradients and they were interpreted as the sum of hydrostatic and wet components. In order to compare to the wet component inferred by the WVR, we subtracted the hydrostatic component computed from the reanalysis product of the European Centre for Medium-Range Weather Forecasts (ECMWF), ERA5, from the total gradient to get the GNSS wet gradient. The hydrostatic gradients at the site are much less variable compared to the wet ones, and especially for time scales of minutes to hours (Elgered et al., 2019).

The data processing was run for three different elevation cutoff angles (3°, 10° and 15°). For each elevation cutoff angle we used four different combinations of GNSS constellations in the processing: GPS only (G), GPS + Glonass (GR), GPS + Galileo (GE), and GPS + Glonass + Galileo (GRE). Due to limitations in the receiver capacity, not all BeiDou observations were recorded. Therefore we decided not to include BeiDou data in our GNSS data processing.

An example of the sky coverage of the observations for different GNSS constellations, applying an elevation cutoff angle of

3°, is shown in Figure 1 for the ONS1 station. The three systems show a similar geometry for the observations. At this latitude a significant part of the sky, just north of the zenith direction, is never sampled. The Glonass satellite orbits have a higher inclination angle implying that a smaller part of the sky is not sampled compared to GPS and Galileo. As a consequence there are more observations from Glonass to the north, especially below the elevation angle of 20°. It is therefore interesting to study how this difference will affect the quality of the estimated gradients.





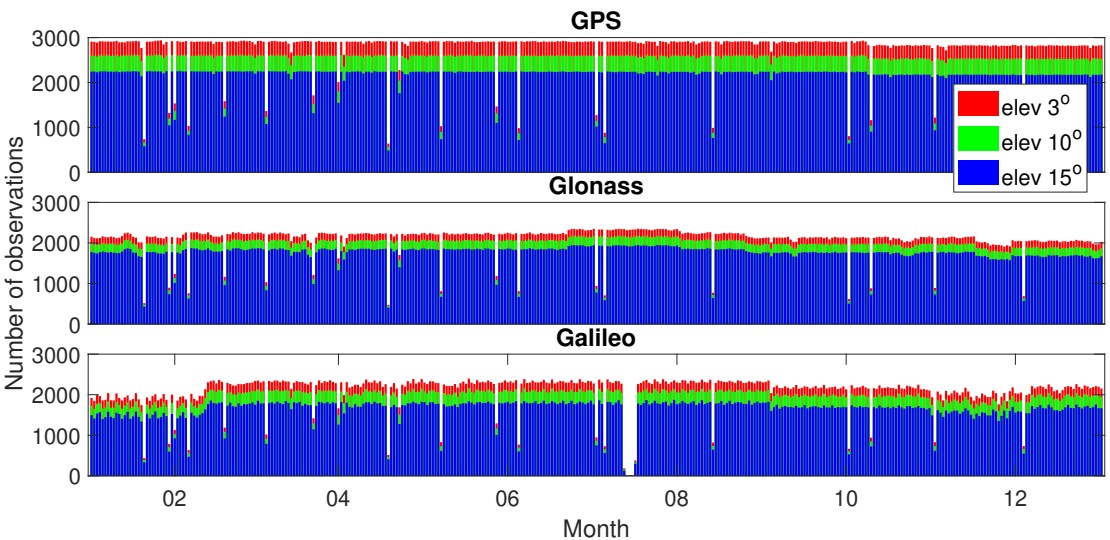

**Figure 2.** Number of daily observations for each GNSS constellation acquired by ONS1 applying three different elevation cutoff angles (3°, 10°, and 15°).

Figure 2 depicts the number of daily observations given for each GNSS constellation and obtained for each elevation cutoff angle. Some days have less or no data due to receiver problems, especially from 13 to 15 July, where no Galileo observations were recorded. Averaged over the year, for the GPS system, the number of observations drops about 11 % and 23 % when the elevation cutoff angle changes from 3° to 10° and 15°, respectively. For the Galileo system, the corresponding decrease in the number of observations are 10 % and 22 %, respectively, while for Glonass, the corresponding values are 8 % and 17 %, respectively.

## 2.2 Water Vapour Radiometer (WVR)

The WVR, shown in Figure 3, is located close to the GNSS sites, 9 m from ONSA and 59 m from ONS1, and with height differences of less than 2 m. The WVR was designed in order to provide independent estimates of the wet propagation delays for space geodetic applications. It measures the sky brightness temperature on and off the water vapour emission line at 22 GHz. More detailed specifications are given by Elgered et al. (2019).

Starting in January 2019 the observations were scheduled in 5 min long cycles with the ambition to sample the whole atmosphere at elevation angles above 25°, which is illustrated in Figure 4. Data were acquired throughout 2019, except from mid August to early October because of a failure caused by a thunder storm.

A four-parameter model was used to estimate the mean ZWD, a linear trend in the ZWD, and east and north linear horizontal gradients over 5 min (Davis et al., 1993). Before the model was applied all observations during rain and with a liquid water





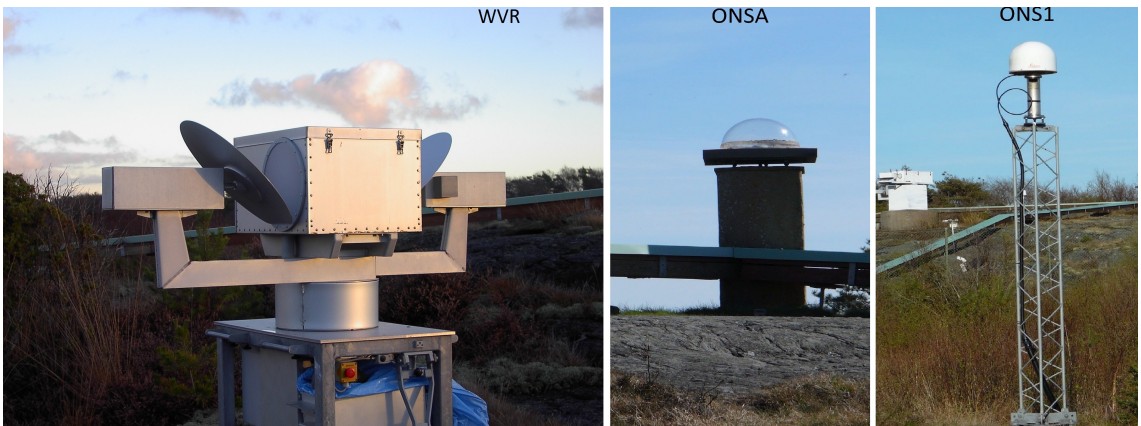

**Figure 3.** The water vapour radiometer (WVR) Konrad and GNSS sites (ONS1 and ONSA) at the Onsala Space Observatory.

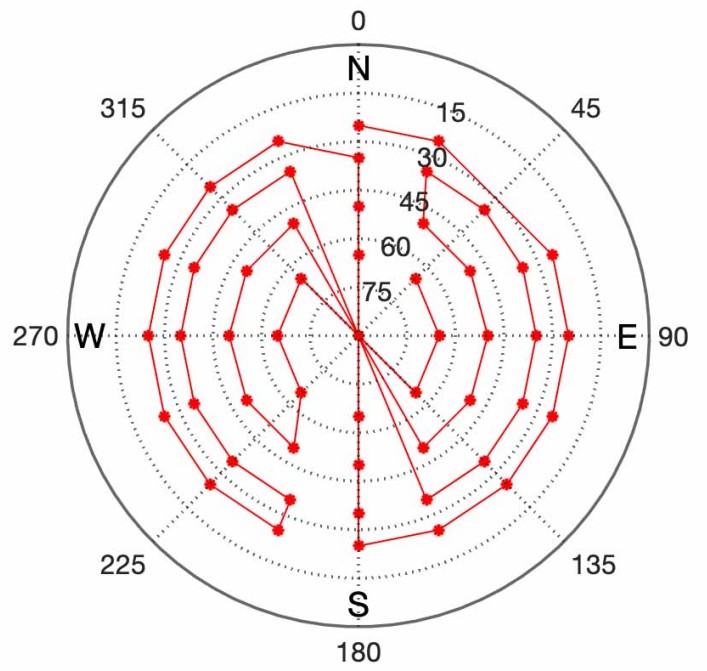

**Figure 4.** One cycle of the WVR observations consists of 52 observations. The cycle is repeated every 5 min. Note that the zenith point is only measured once, the first time it passes over zenith.

content larger than 0.7 mm were removed. Thereafter, the quality of the data was assessed through manual editing. Gain jumps occurring sometimes at the beginning of a 5 min cycle were frequent during the whole year. When such a jump was identified one or several complete 5 min cycles were removed. Thereafter, when the model was applied, we required that at least 40 of the





52 observations were available in the 5 min cycle period. Thereby we eliminated the possibility that the large gain jumps could

have an impact on the estimated ZWD trends and gradients for the 5 min cycle since it is not synchronised with the estimation period. Smaller gain jumps may still have degraded the accuracy of the estimated gradients. An overall reason for applying this strict editing was that the primary goal was to have accurate gradients from the WVR rather than a statistical characterization of the specific atmospheric conditions at the site.

Finally it is noted that the WVR estimates are completely independent of the corresponding estimates from the GNSS data.

The estimated ZWD from the WVR data is shown in Figure 5. The seasonal dependence is clearly visible as well as a large short term variability.

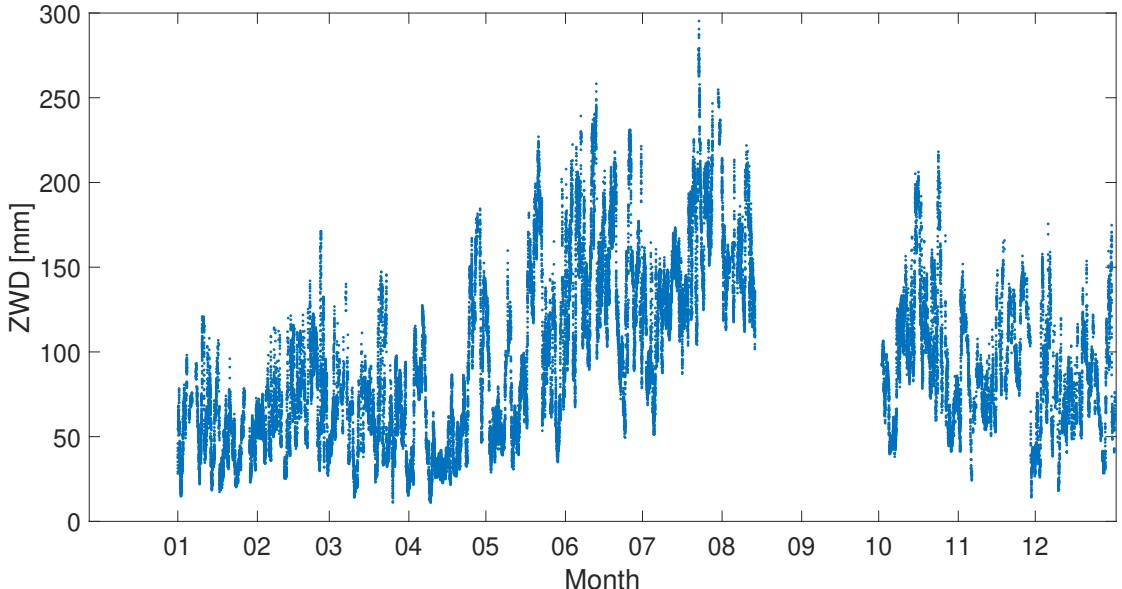

**Figure 5.** The time series of the ZWD estimated from the WVR data for each 5 min cycle. The radiometer was repaired from mid August to early October.

## 3 Results

### 3.1 Estimated gradients and their formal errors

Before carrying out comparisons of the gradients estimated from different GNSS solutions with the WVR gradients we in-

110 vestigate the characteristics of the input data. Table 1 summarizes the statistics of estimated gradients and their corresponding formal errors. A few features of the data are worth noting. The mean and the variability (standard deviation) of the estimated gradient amplitude increase with increasing elevation cutoff angle together with its mean formal error. The gradient amplitudes estimated by the WVR are approximately twice as large as the GNSS gradient amplitudes at 3° cutoff angle but the decreases





**Table 1.** The mean and standard deviations (SD) of the east and the north gradients, and the gradient amplitude, together with the mean and SD of their 1-sigma formal errors.

| GNSS constellation with elevation cutoff angle c.f. the WVR | East, $\Xi_e$ | | | | North, $\Xi_n$ | | | | Amplitude, $\sqrt{\Xi_e^2 + \Xi_n^2}$ | | | |
|---|---|---|---|---|---|---|---|---|---|---|---|---|
| | Estimated | | Formal error | | Estimated | | Formal error | | Estimated | | Formal error | |
| | Mean | SD | Mean | SD | Mean | SD | Mean | SD | Mean | SD | Mean | SD |
| | (mm) | (mm) | (mm) | (mm) | (mm) | (mm) | (mm) | (mm) | (mm) | (mm) | (mm) | (mm) |
| G(PS) 3° | 0.00 | 0.43 | 0.14 | 0.04 | 0.09 | 0.40 | 0.14 | 0.04 | 0.48 | 0.36 | 0.20 | 0.05 |
| GE[a] 3° | −0.00 | 0.45 | 0.10 | 0.03 | 0.10 | 0.41 | 0.10 | 0.03 | 0.49 | 0.38 | 0.15 | 0.03 |
| GR[b] 3° | −0.00 | 0.45 | 0.11 | 0.03 | 0.10 | 0.40 | 0.10 | 0.02 | 0.48 | 0.37 | 0.15 | 0.03 |
| GRE[c] 3° | −0.00 | 0.45 | 0.09 | 0.02 | 0.10 | 0.41 | 0.09 | 0.02 | 0.49 | 0.38 | 0.13 | 0.03 |
| G(PS) 10° | −0.05 | 0.48 | 0.22 | 0.03 | 0.04 | 0.48 | 0.22 | 0.03 | 0.56 | 0.38 | 0.31 | 0.05 |
| GE[a] 10° | −0.02 | 0.52 | 0.17 | 0.03 | 0.04 | 0.50 | 0.18 | 0.03 | 0.58 | 0.42 | 0.25 | 0.04 |
| GR[b] 10° | −0.02 | 0.51 | 0.17 | 0.02 | 0.05 | 0.49 | 0.17 | 0.02 | 0.58 | 0.41 | 0.25 | 0.03 |
| GRE[c] 10° | −0.02 | 0.53 | 0.15 | 0.02 | 0.03 | 0.51 | 0.15 | 0.02 | 0.59 | 0.44 | 0.22 | 0.03 |
| G(PS) 15° | −0.06 | 0.48 | 0.33 | 0.04 | −0.03 | 0.60 | 0.33 | 0.05 | 0.66 | 0.40 | 0.47 | 0.06 |
| GE[a] 15° | −0.02 | 0.51 | 0.26 | 0.04 | −0.07 | 0.61 | 0.27 | 0.04 | 0.68 | 0.41 | 0.38 | 0.05 |
| GR[b] 15° | 0.00 | 0.50 | 0.26 | 0.03 | −0.05 | 0.60 | 0.26 | 0.03 | 0.67 | 0.40 | 0.37 | 0.04 |
| GRE[c] 15° | 0.00 | 0.53 | 0.23 | 0.03 | −0.09 | 0.61 | 0.23 | 0.03 | 0.69 | 0.43 | 0.33 | 0.04 |
| WVR | 0.12 | 1.00 | 0.09 | 0.08 | 0.19 | 0.80 | 0.08 | 0.07 | 0.99 | 0.84 | 0.12 | 0.11 |

[a] GPS + Galileo;  [b] GPS + Glonass;  [c] GPS + Glonass + Galileo

to around 50 % as large for the cutoff angle of 15°. The GNSS gradient amplitudes are about twice as large as their formal errors. The WVR gradient amplitudes are about eight times larger than their mean formal errors, but the variability of the WVR formal errors is significantly larger than those from GNSS. This is due to a varying uncertainty in the measured sky brightness temperatures. These variations are taken into account in the following comparisons by using of the formal errors of the GNSS and the WVR gradients when calculating the weighted root-mean-square (WRMS) differences and correlations.

### 3.2 Comparison of gradients from GNSS and WVR

We first carry out comparisons of the gradients estimated from the different GNSS constellations and using the three different elevation cutoff angles, presented in Table 1, with the WVR gradients. Even though the gradients estimated from both the GNSS and the WVR data have a temporal resolution of 5 min, the estimates are not centered at exactly the same time epochs. It is therefore necessary to synchronize the time series to compare the gradients. We first present results with the highest available temporal resolution of 5 min where the WVR gradients were interpolated to the epochs in the GNSS time series using the temporal Gaussian filter as described by Ning et al. (2012) with a full width at half maximum (FWHM) of ± 2.5 min.

Table 2 shows the WRMS differences and correlations of the east and the north gradients for the whole year of 2019. As expected, using the data from multi-GNSS, we note a significant improvement (an increase in the correlation of up to 20 % and a maximum reduction of the WRMS difference of 11 %). Our interpretation is that the geometry of the observations is improved when observations from additional systems are added, especially in the south-north direction (see Figure 1). We also note that the GRE solution, in general, gives the best agreement with the WVR gradients.





**Table 2.** The WRMS differences and correlations of the east and north gradients, obtained from different satellite constellations, relative to the WVR data.

| GNSS solution | G(PS) | | GE[a] | | GR[b] | | GRE[c] | |
|---|---|---|---|---|---|---|---|---|
| + cutoff angle | East | North | East | North | East | North | East | North |
| WRMS differences | (mm) | (mm) | (mm) | (mm) | (mm) | (mm) | (mm) | (mm) |
| ONS1 3° | 0.70 | 0.56 | 0.67 | 0.52 | 0.67 | 0.52 | 0.66 | 0.50 |
| ONS1 10° | 0.74 | 0.63 | 0.69 | 0.59 | 0.70 | 0.58 | 0.67 | 0.56 |
| ONS1 15° | 0.80 | 0.79 | 0.75 | 0.78 | 0.75 | 0.76 | 0.72 | 0.76 |
| Correlation coefficients | | | | | | | | |
| ONS1 3° | 0.62 | 0.61 | 0.63 | 0.64 | 0.64 | 0.65 | 0.63 | 0.66 |
| ONS1 10° | 0.61 | 0.55 | 0.66 | 0.61 | 0.65 | 0.63 | 0.68 | 0.66 |
| ONS1 15° | 0.55 | 0.37 | 0.61 | 0.41 | 0.60 | 0.43 | 0.63 | 0.46 |

[a] GPS + Galileo;   [b] GPS + Glonass;   [c] GPS + Glonass + Galileo

For the GPS-only solution, the best correlation is obtained for the elevation cutoff angle of 3°, especially for the north gradients. This is however not the case for the multi-GNSS solutions (GR, GE, and GRE) where the best correlation for the east gradient is obtained for the elevation cutoff angle of 10°. For the GRE solutions, the correlations given by the 3° and 10° solutions are similar for the north gradients. These results indicate that the choice of elevation cutoff angle is a compromise between having a good geometry and avoiding elevation-angle-dependent systematic errors, e.g., multipath effects. Related to this is the error introduced by the assumption that the turbulent atmosphere may be modelled with just a linear gradient when the elevation cutoff angle of 25° has to be used for the WVR observations in order to avoid ground noise pickup. As was presented in Table 1 the estimated size of the gradients was in general increasing when using a decreasing sky coverage of the observations.

In spite of that the WVR and the GNSS sample different parts of the sky it is noted that the agreement becomes worse for all GNSS solutions when the 15° elevation cutoff angle is used. Our interpretation is that it is because many important observations are removed, especially in the north direction. We also note that even though there are slightly more observations from the Glonass system contributing to the south-north direction, especially below the elevation angle of 20° (see Figure 1), the GR solution does not give a significantly better agreement with the WVR than the one for the GE solution.

In order to study any seasonal variability we compare the estimated gradients from the GNSS and the WVR for each month. The GNSS gradients are obtained using different constellations and a cutoff angle of 3° (see Figure 6). The change in the correlation is large from month to month and these changes seem to be related to the amplitude of the ZWD (see Figure 5). In general, a large ZWD variability results in a larger dynamic range for the gradients and consequently also a larger correlation. Figure 6 shows higher gradient correlations for June and July which is consistent with the results given by Elgered et al. (2019). To investigate how well the GNSS data capture large gradients, we carried out the same comparison using data from June and July only. Another reason for focusing on these two months is to include the ONSA GNSS station for validation purposes. Multi-GNSS data from ONSA started to be acquired only in April 2019.

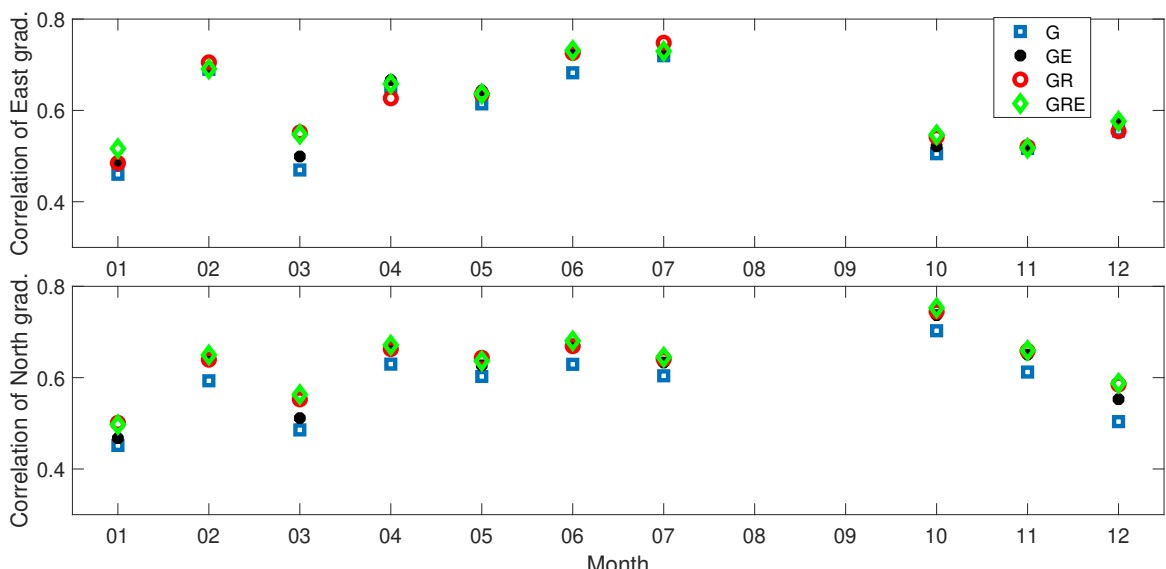

**Figure 6.** Correlations between estimated gradients from the GNSS and the WVR data calculated for each month.

The results are summarized in Table 3. Because of the larger gradients the WRMS differences increase in all cases. However, at the same time we see higher correlations between GNSS and WVR gradients compared to the whole dataset (see Table 2).

The two GNSS stations (ONS1 and ONSA) show very similar agreements with the WVR gradients. This is partly expected since they are located close to each other and therefore the gradients from the two sites are estimated based on the same observational directions and are affected by common error sources, such as orbit errors. However, it is of interest to note that the different antenna mountings (see Figure 3) do not have a significantly different impact on the estimated gradients. Therefore, we continue to use only ONS1 GNSS data in the following.



**Table 3.** The WRMS differences and correlations of the east and the north gradients, obtained from different satellite constellations, relative to the WVR data for June and July 2019.

| GNSS station | G(PS) | | GE[a] | | GR[b] | | GRE[c] | |
|---|---|---|---|---|---|---|---|---|
| + cutoff angle | East | North | East | North | East | North | East | North |
| WRMS differences | (mm) | (mm) | (mm) | (mm) | (mm) | (mm) | (mm) | (mm) |
| ONS1 3° | 0.82 | 0.67 | 0.79 | 0.62 | 0.78 | 0.62 | 0.78 | 0.61 |
| ONSA 3° | 0.82 | 0.65 | 0.79 | 0.62 | 0.78 | 0.61 | 0.77 | 0.60 |
| ONS1 10° | 0.83 | 0.71 | 0.77 | 0.66 | 0.77 | 0.65 | 0.73 | 0.63 |
| ONSA 10° | 0.82 | 0.70 | 0.75 | 0.65 | 0.75 | 0.64 | 0.72 | 0.63 |
| ONS1 15° | 0.91 | 0.88 | 0.83 | 0.85 | 0.83 | 0.83 | 0.79 | 0.83 |
| ONSA 15° | 0.92 | 0.85 | 0.84 | 0.80 | 0.83 | 0.80 | 0.79 | 0.78 |
| Correlation coefficients | | | | | | | | |
| ONS1 3° | 0.71 | 0.62 | 0.73 | 0.65 | 0.75 | 0.66 | 0.74 | 0.67 |
| ONSA 3° | 0.70 | 0.66 | 0.73 | 0.68 | 0.74 | 0.70 | 0.74 | 0.69 |
| ONS1 10° | 0.73 | 0.59 | 0.77 | 0.66 | 0.78 | 0.67 | 0.80 | 0.70 |
| ONSA 10° | 0.73 | 0.61 | 0.78 | 0.66 | 0.79 | 0.68 | 0.81 | 0.70 |
| ONS1 15° | 0.68 | 0.41 | 0.73 | 0.46 | 0.75 | 0.49 | 0.77 | 0.52 |
| ONSA 15° | 0.66 | 0.43 | 0.74 | 0.49 | 0.75 | 0.50 | 0.77 | 0.53 |

[a] GPS + Galileo;  [b] GPS + Glonass;  [c] GPS + Glonass + Galileo


## 3.3 Effective temporal resolutions from 5 min to 24 h

As mentioned in the introduction GNSS gradients have been compared to other independent estimates over different time scales and temporal resolutions. We therefore averaged the GNSS and the WVR gradients by applying a gaussian window with different FWHM, from $\pm$ 2.5 min to $\pm$ 720 min for further comparisons. As described in Section 3.2, when a FWHM of $\pm$ 2.5 min is used, only the WVR gradients were interpolated to the epochs in the GNSS time series. For all FWHM larger than $\pm$ 2.5 min both the GNSS and the WVR gradients are interpolated to the epochs at 0, 5, 10... 55 min after the hour. The requirement to calculate a value at a specific epoch is that at least half of the original data points (with a 5 min resolution) exist within the FWHM. In the following we will refer to and use the FWHM as an effective temporal resolution, $\Delta t_{\text{eff}}$, from 5 min to 1440 min (one day), although the time series will still have a value every 5 min.

The resulting WRMS differences and correlations are shown in Figure 7. It is clear that the WRMS differences decrease when the $\Delta t_{\text{eff}}$ increases and more variations of the gradients are averaged out. Over all GNSS solutions, the highest correlation for the east gradients is obtained when a $\Delta t_{\text{eff}}$ of 2 h is used, while for the north gradients, the best agreement is seen for a $\Delta t_{\text{eff}}$ of 6 h. Figures 8 and 9 depict the gradients estimated from the GNSS data and the GRE solution for a 3° elevation cutoff angle, against the gradients obtained from the WVR data for four different values for $\Delta t_{\text{eff}}$ (5 min, 2 h, 12 h, and 24 h). It illustrates that even when the gradients are averaged over one day, i.e., applying $\Delta t_{\text{eff}}$ of 24 h, there are substantial variations left which still give a clear correlation between the GNSS and the WVR data.

We also study a specific event of short lived gradients, associated with rapid changes in the ZWD, starting from 0 h, 23 July (see Figure 10). There is a passage of a warm front, indicated by a sudden increase in the ZWD, during the late evening of 24 July. As a result, we see a large gradient towards the west direction. They are detected by both GNSS and WVR but the amplitudes of the gradients from GNSS are much smaller. For the north direction, there are also some large gradients, i.e., at 8 and 17 h of 23 July, which are detected by both GNSS and WVR. Also in this case the amplitudes from GNSS are smaller. In addition, the multi-GNSS solutions have a slightly higher possibility to capture sudden short lived gradients.

The results from the comparisons using different $\Delta t_{\text{eff}}$ indicate that due to the poor geometry of the GNSS observations (especially in the south-north direction), if the gradient happens suddenly, the GNSS data do not capture the full picture of the gradients. When a small $\Delta t_{\text{eff}}$ is used, all gradients are kept including the ones which are not correctly detected by the GNSS data. As a result, the correlation between the GNSS and the WVR data is deteriorated and this is the case when a $\Delta t_{\text{eff}}$ of 5 min is applied. However, when $\Delta t_{\text{eff}}$ is too large (i.e., 24 h), the larger gradients which are captured correctly by both the GNSS and the WVR data will also be averaged out. The range of variations and the correlation decrease.

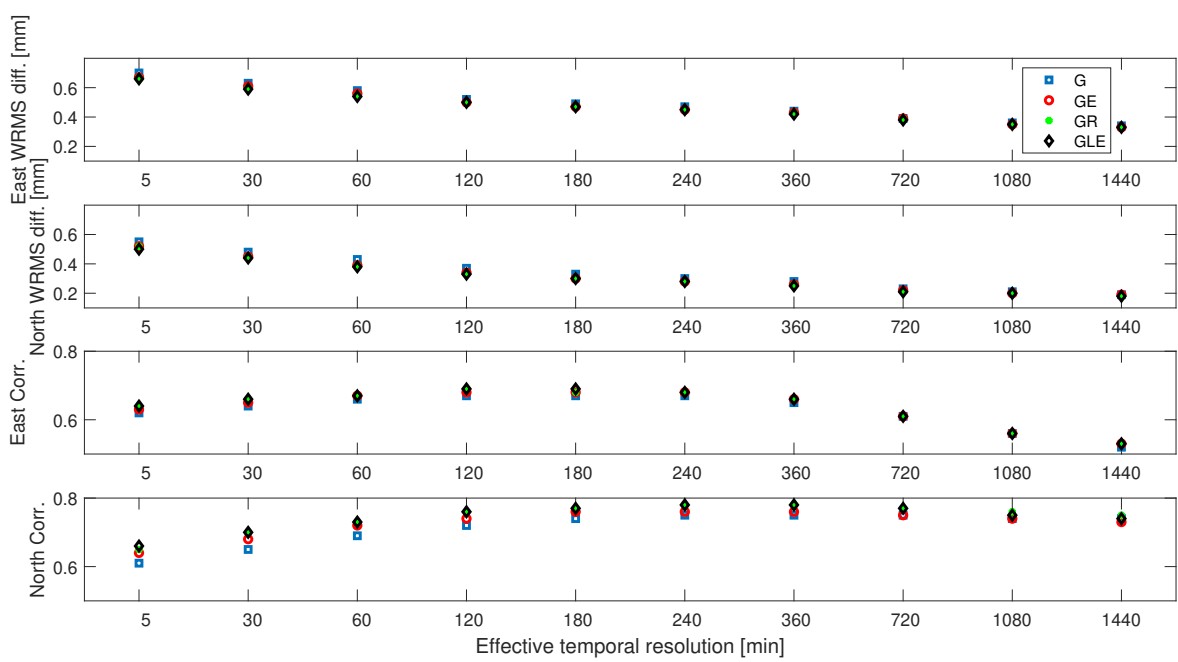

**Figure 7.** Correlations and WRMS differences between estimated gradients from the GNSS and the WVR data using different effective temporal resolutions, $\Delta t_{\text{eff}}$.

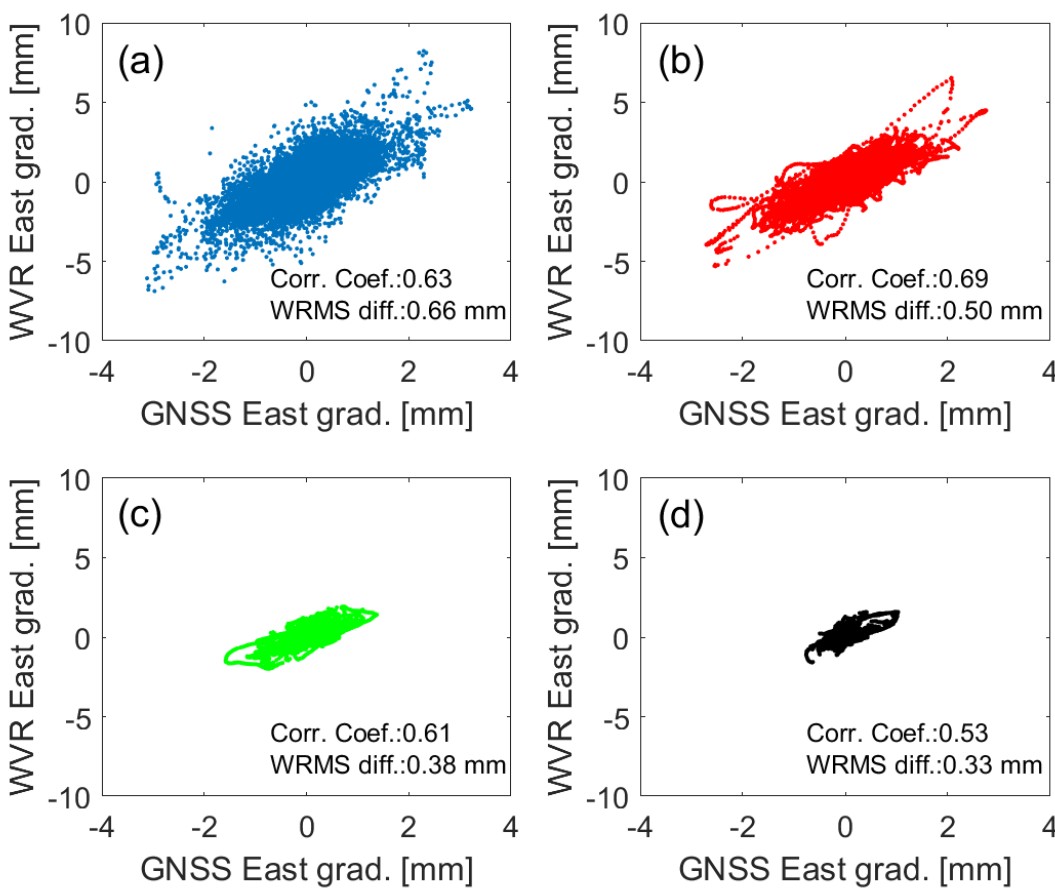

**Figure 8.** Correlations between estimated east gradients from the GNSS, given by the GRE solution, and the WVR data. Four different vaules of $\Delta t_{\mathrm{eff}}$ (a) 5 min, (b) 2 h, (c) 12 h, and (d) 24 h are used.



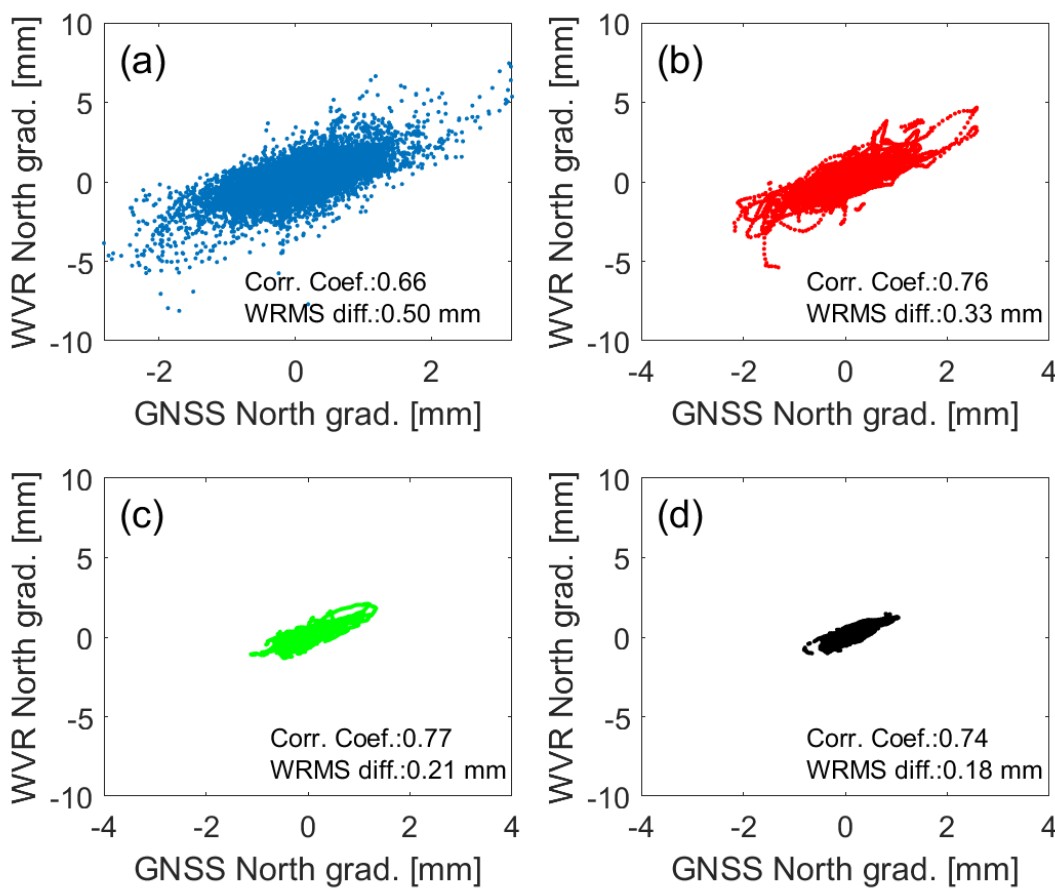

**Figure 9.** Correlations between estimated north gradients from the GNSS, given by the GRE solution, and the WVR data. Four different values of $\Delta t_{\text{eff}}$ (a) 5 min, (b) 2 h, (c) 12 h, and (d) 24 h are used.





**Figure 10.** ZWD (top) and gradient time series from the WVR and the GNSS solution using an elevation cutoff angle of 3°. From the second to the bottom panel, east and north gradients are shown for $\Delta t_{\mathrm{eff}}$ of 5 min, 2 h, and 24 h, respectively. Note the different scales for the east and the north gradient graphs.





**Table 4.** The changes in WRMS differences and correlations of east and north gradients when a weaker constraint is applied in the GNSS data processing. The changes are the results from the constraint of $1.0 \, \mathrm{mm}\sqrt{\mathrm{h}}^{-1}$ relative to the $0.3 \, \mathrm{mm}\sqrt{\mathrm{h}}^{-1}$ (in Table 3).

| | G(PS) | | GE[a] | | GR[b] | | GRE[c] | |
|---|---|---|---|---|---|---|---|---|
| | East | North | East | North | East | North | East | North |
| WRMS differences | (mm) | (mm) | (mm) | (mm) | (mm) | (mm) | (mm) | (mm) |
| 3° | 0.01 | 0.03 | 0.03 | 0.04 | 0.04 | 0.04 | 0.03 | 0.04 |
| 10° | −0.04 | 0.02 | −0.03 | 0.01 | −0.03 | 0.00 | −0.02 | 0.01 |
| Correlation coefficients | | | | | | | | |
| 3° | 0.02 | 0.01 | 0.00 | 0.01 | −0.01 | 0.00 | 0.00 | −0.01 |
| 10° | 0.03 | 0.04 | 0.03 | 0.03 | 0.02 | 0.04 | 0.01 | 0.03 |

[a] GPS + Galileo;  [b] GPS + Glonass;  [c] GPS + Glonass + Galileo

## 3.4 Different constraints for gradient variability

All GNSS-derived gradients were so far estimated using a random walk model with a constraint value of $0.3 \, \mathrm{mm}\sqrt{\mathrm{h}}^{-1}$ (see
Section 2.1). This value may be too small, especially when we have more observations from multi-GNSS constellations, to allow the GNSS data to detect sudden large gradients. In order to investigate this issue, we have processed the GNSS data from June and July again with the 5 min temporal resolution, applying a weaker constraint of $1.0 \, \mathrm{mm}\sqrt{\mathrm{h}}^{-1}$. The changes in WRMS differences and correlations, relative to the solution using the constraint value of $0.3 \, \mathrm{mm}\sqrt{\mathrm{h}}^{-1}$, are shown in Table 4 where the GNSS gradients are estimated applying the elevation cutoff angles of 3° and 10°. The formal errors obtained when
using the weak constraint of $1.0 \, \mathrm{mm}\sqrt{\mathrm{h}}^{-1}$ is about twice as large compared to the ones obtained when using the constraint of $0.3 \, \mathrm{mm}\sqrt{\mathrm{h}}^{-1}$.

When a weak constraint is applied with an elevation cutoff angle of 3°, the WRMS differences increase for both the east and the north gradients while the correlations are almost the same. We note that for the 10° solution, the east gradients obtained from using the weak constraint values result in smaller WRMS differences compared to the WVR gradients. A slight improvement
is also seen in the correlations for both the east and the north gradients. We interpret the result as the compromise between capturing large gradients and including more noise. When a low elevation cutoff angle is used, the GNSS measurements will be more sensitive to the noise from the environment, i.e., multipath effects. In addition, the GNSS signals will have a lower signal-to-noise ratio due to the longer path through the atmosphere. When a weak constraint is applied, individual observations will have a larger influence on the estimated gradients.
More details are seen in Figure 11, depicting the time series of the gradients for the two and a half days, starting at 0 h, 23 July, from the WVR and the GNSS data based on the GRE solutions. There are several peaks, i.e., large gradients, shown for these two and a half days, i.e., east gradients at 21 h of 24 July and north gradients at 8 and 17 h of 23 July. When a weak constraint is applied, there is a clear improvement in tracking those larger gradients when the elevation cutoff angle of 10° is used. This is not the case when the lower elevation cutoff angle of 3° is used, possibly because the sampled atmosphere is more
different compared to that observed by the WVR.

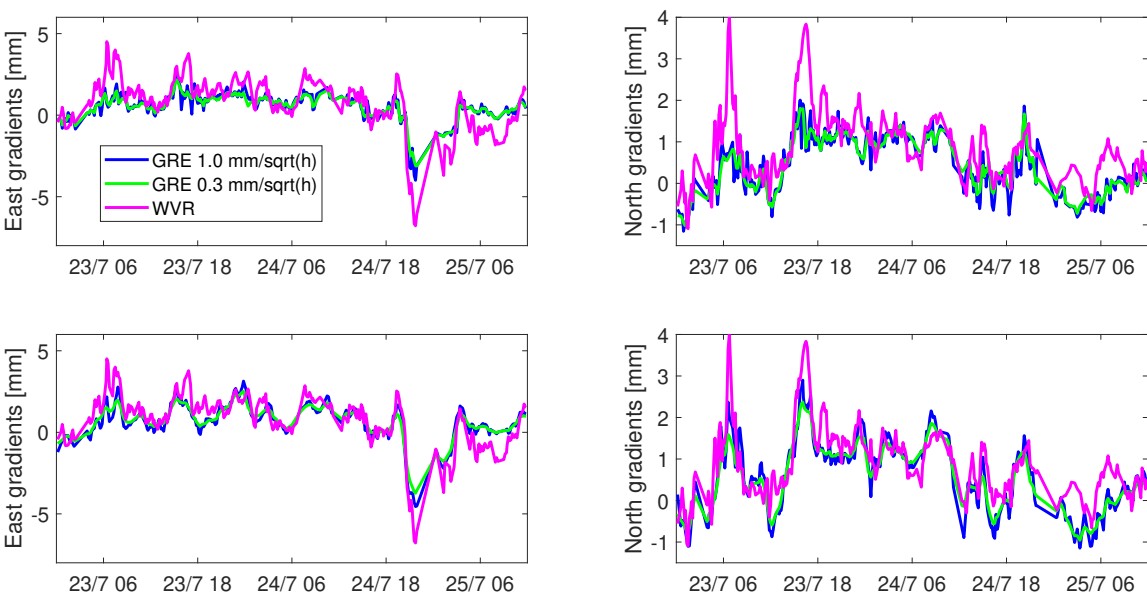

**Figure 11.** Gradient time series from the WVR and from GNSS (GRE) applying two different constraint values (0.3 and 1.0 $\text{mm}\sqrt{\text{h}}^{-1}$) using elevation cutoff angles of $3°$ (upper panel) and $10°$ (lower panel).



## 4 Conclusions

We have estimated linear horizontal gradients using one year of data acquired from the GNSS site ONS1 located on the Swedish west coast. The GNSS-derived gradients were compared to the ones obtained from a collocated WVR. Overall the multi-GNSS solutions, i.e., combinations of GPS, Glonass, and Galileo, show small but significant improvements with the WVR gradients compared to the GPS-only solution (Tables 2 and 3).

For the GPS-only solution, the best agreement, in terms of the correlation coefficient with the WVR gradients, is obtained when using an elevation cutoff angle of $3°$. For the multi-GNSS solution using all three constellations, the best agreement with the WVR data is obtained for the solution with an elevation cutoff angle of $10°$. The difference is largest for the east component which has the better sky coverage (Figure 1 and Tables 2 and 3). This indicates that if there are a sufficient number of observations, the low elevation observations are not that important. This is especially true when the comparison is made to a WVR using observations evenly spread over the sky above an elevation angle of $25°$. It is also an indication that a linear model for horizontal variations in the wet refractivity does not describe the turbulent atmosphere well during all conditions.

We investigated different effective temporal resolutions, $\Delta t_{\mathrm{eff}}$, of the compared time series. For all GNSS solutions, the highest correlations obtained for the east and the north gradients are for a $\Delta t_{\mathrm{eff}}$ of 2 h and 6 h, respectively (Figure 7). When these $\Delta t_{\mathrm{eff}}$ are applied, strong gradients of short duration detected by the WVR, but not by GNSS, are averaged out and as a result the correlation increases. When estimating GNSS gradients the choice of $\Delta t_{\mathrm{eff}}$ is a compromise between getting a high correlation and loosing track of rapid gradient variations. However, when $\Delta t_{\mathrm{eff}}$ is even larger, e.g., 24 h, all gradients are further averaged and the dynamic range of gradient size and the correlation decreases (Figures 8, 9, and 10).

Furthermore, weakening the constraint used when estimating the GNSS gradients from $0.3\,\mathrm{mm}\sqrt{\mathrm{h}}^{-1}$ to $1.0\,\mathrm{mm}\sqrt{\mathrm{h}}^{-1}$ helps the GNSS data to track short-lived gradients, approaching a time scale of 5 min, however at the cost of increased formal errors (Table 4 and Figure 11).

Possible improvements to study in similar future work would be to include BeiDou observations and use a WVR with a better stability. It would also be of interest to carry out a similar study at low latitude GNSS sites where the sky coverage is better and perhaps also the atmosphere is more variable.

*Acknowledgement.* We thank Tobias Nilsson for providing the ZHD gradients from the ERA5 data.

*Data availability.* The input GNSS data, in RINEX format, are available from EUREF, https://igs.bkg.bund.de/dataandproducts/browse. The ERA5 data are accessible from https://rda.ucar.edu/datasets/ds633.0/. The estimated gradients from the GNSS and the WVR data will be registered and archived by the Swedish National Data Service (SND) if and when the paper is accepted for publication. Before that these data are made available by the authors.



*Author contributions.*   The two authors (TN and GE) planned the work and the structure of the paper together. TN performed the GNSS data analyses and GE performed the WVR data analyses. Both contributed to the writing of the manuscript and approved it before the submission.

*Competing interests.*   The authors declare that they have no conflict of interest.





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
