# Peer review of "High temporal resolution wet delay gradients estimated from multi-GNSS and microwave radiometer observations"

_Atmospheric Measurement Techniques, 2021_

## Referee Comment (RC1)

**Manuscript Review**

Title: High temporal resolution wet delay gradients estimated from multi-GNSS and

microwave radiometer observations

Authors: Tong Ning and Gunnar Elgered Journal: Atmospheric Measurement Techniques

This manuscript compares the linear horizontal wet delay gradients estimated from multi-GNSS against the corresponding values estimated from WVR using one year data. It is shown that the improvement obtained for the solution using multi-GNSS data is obvious, i.e. an increase in the correlation coefficient of 11 % for the east gradient and 20 % for the north gradient (by comparing with those from the GPS-only solution). In addition, the authors also investigate the impact of different temporal resolutions from 5 min to one day, and the result indicates that the highest correlations obtained for the east and north gradients are for a resolution of 2 h and 6 h, respectively. And for a resolution of one day, all gradients are averaged and the dynamic range of gradient size and the correlation decreases. In addition, it is also shown that, for catching the gradients with a sudden occurrence nature, the GNSS underperforms the WVR. The paper in general is well-written and gives detailed information about the related processes. I have only a few comments for the authors:

**Main comments:**

- 1. Section 2.2: How to identify the jumps occurring sometimes at the beginning of the 5 min cycle when you estimate the gradients using WVR. And how many observations from WVR were removed or how many observations are used to compare with the gradients estimated from GNSS?
- 2. Line 111-112: It is very obvious that the mean and variability of the estimated gradients amplitude increase with an increase in elevation cutoff angle together with its formal error. Why is this?
- 3. Line 112-114: "The gradient amplitudes estimated by the WVR are approximately twice as large as the GNSS gradient amplitudes at 3° cutoff angle but the decreases to around 50% as large for the cutoff angle of 15°". I do not find the feature from Table 1. Which column of the table can be used to reach the conclusion?
- 4. The formal errors of WVR are larger than ones of GNSS in Table 1. Also as is mentioned in line 116-117, the uncertainty in measured sky brightness is unstable. It seems that the uncertainty of WVR is larger than that of GNSS. Therefore, are the comparison results credible?

---

## Referee Comment (RC2)

Review: High temporal resolution wet delay gradients estimated from multi-GNSS and microwave radiometer observations

The study analyzes wet delay gradients from the gnss and the water vapor radiometer (wvr). Results are consistent with results from previous studies. Some new interesting details are provided. In general i find the paper well written and i suggest publication after the following points are addressed:

P 2, L 36: "...The temporal resolutions of such comparisons are to our knowledge so far limited to 1 h for WVRs (Lu et al., 2016), 2 h for VLBI (Steigenberger et al., 2007), and 6 h for numerical weather models (Zus et al., 2019)." I think at this point it is worth to mention the following recent paper:

Kačmařík, M., Douša, J., Zus, F., Václavovic, P., Balidakis, K., Dick, G., and Wickert, J.: Sensitivity of GNSS tropospheric gradients to processing options, Ann. Geophys., 37, 429–446, https://doi.org/10.5194/angeo-37-429-2019, 2019.

In this paper hourly nwm/gnss data were utilized in the comparisons.

P2, L55: "...We used the Vienna Mapping Function 1 (VMF1) (Boehm et al.,552006) to map the zenith delay and the gradient mapping function was the one presented by Bar-Sever et al. (1998)" Please provide the formulas for the wet delay gradients somewhere in the manuscript. Maybe this is the right point to do so. For many readers it is not clear what you mean by wet delay gradients. Also the formulas will help to understand why the estimated wet delay gradients depend on the observation geometry, elevation cut off angle, elevation dependent weighting etc.

P3, L65: "...In order to compare to the wet component inferred by the WVR, we subtracted the hydrostatic component computed from the reanalysis product of the European Centrefor Medium-Range Weather Forecasts (ECMWF), ERA5, from the total gradient to get the GNSS wet gradient…" Please provide the details (or reference) on how you calculated the hydrostatic gradient from the weather model. What about the wet gradient? Is it worth to be included in your comparisons?I mean is it close to the wvr and the gnss estimates?

P4, L95: You make use of a four parameter model to obtain tropospheric parameters. This is very similar to what is done with the gnss. Why do you not apply similar constraints?

P5: In figure 3 you show the gnss installation. Can you provide a multipath map for the two installations? What do you think is the main limiting factor for the accuracy of the gnss gradient estimates?

P5: In figure 4 you show the observation geometry that is used to estimate the gradients. With such measurements you could investigate the role of the observation geometry, right? For example, you could remove certain observations to the north to get an idea on how well the gradient can be estimated with the limited observation geometry of the gnss.

Figure 10: Do you have an explanation for the significant bias in the zwd? Out of curiousity: did you compare the measured wvr zwd (measured in zenith direction) with the estimated wvr zwd (estimated by least square adjustment)?

Figure 11: This is a very interesting plot. again, my question is the following: what would happen if you apply constraints in the wvr estimation procedure.

---

## Author Comment (AC1)

Dear Referee #1,

We appreciate your questions and suggestions for improvements. They have been adopted and/or further discussed as described in the following. All responses are given in italic and green font.

Sincerely yours

T. Ning and G. Elgered

Main comments:

Section 2.2: How to identify the jumps occurring sometimes at the beginning of the 5 min cycle when you estimate the gradients using WVR. And how many observations from WVR were removed or how many observations are used to compare with the gradients estimated from GNSS?

*Response: We have added the following details describing how the WVR data were prepared for the GNSS comparison:*

"The jumps were identified by viewing the ZWD during each day. The temporal resolution is then sufficient to identify a 5 min long group of data that is discontinuous to the adjacent 5 min periods. The jumps were later found to be caused by vibrations when the mechanical waveguide switch was activated at the beginning of each 5 min period. After removing WVR data acquired during rain, that were unstable because of the jumps, and all 5 min periods where there were less than 40 (of the 52 scheduled) observations (typically caused by large liquid water content) we ended up with 56,612 data points. There are 105,120 possible data points in one year. There were 14,236 periods of 5 min during the time when the WVR was in the lab, meaning that the gradients estimated from 62 % of the time when the WVR was operated were used to compare to the ones from GNSS."

Line 111-112: It is very obvious that the mean and variability of the estimated gradients amplitude increase with an increase in elevation cutoff angle together with its formal error. Why is this?

Response: In order to explain the issue, we have added the following text in the updated manuscript: "When the elevation cutoff angle increases, fewer number of observations were included for the gradient estimation. For GNSS, the geometry of the satellite constellation is also deteriorated for a larger elevation cutoff angle. As a result, the formal error of the estimated gradient increases as well as the variability. In addition, when using a lower elevation, the larger volume sensed by GNSS introduces an averaging effect that reduces the mean amplitude of the estimated gradients (see Elgered et al. (2019)")."

Line 112-114: "The gradient amplitudes estimated by the WVR are approximately twice as large as the GNSS gradient amplitudes at 3° cutoff angle but the decreases to around 50% as large for the cutoff angle of 15°". I do not find the feature from Table 1. Which column of the table can be used to reach the conclusion?

Response: In the updated manuscript we have modified the sentence to "As indicated by column 9 in Table 1, the gradient amplitudes estimated by the WVR (0.99 mm) are approximately twice as large as the GNSS gradient amplitudes at 3° cutoff angle, i.e., 0.49 mm for the GRE solution, but they decrease to around 50 % as large for the cutoff angle of 15°, i.e., 0.69 mm for the GRE solution."

The formal errors of WVR are larger than ones of GNSS in Table 1. Also as is mentioned in line 116-117, the uncertainty in measured sky brightness is unstable. It seems that the uncertainty of WVR is larger than that of GNSS. Therefore, are the comparison results credible?

Response: This explanation is added when discussing Table 1:

"The uncertainty of the WVR gradients are scaled meaning that if the true wet delays in the different directions have deviations from the linear gradient model the uncertainties increase.

Such deviations will be common during convection processes and the assumption of linear changes of the wet refractivity in a layered atmosphere will not be accurate. The gradient uncertainty given by GipsyX is not scaled. Therefore, these uncertainties are likely smaller than realistic values."

Related to these comments we are also motivated to add the following text at the end of Section 2.2 describing the WVR data:

"The study does not need to assume that the WVR gradients are more accurate compared to the GNSS ones. The main advantage of the WVR gradients is that they are independent and by comparing these to the gradients from different GNSS solutions we can assess the different GNSS processing methods. Furthermore, since we want to study the agreement with as high temporal resolution as possible, we do not apply constraints to the individual 5 min gradients in order to have them independent from adjacent estimates in terms of the atmospheric signals."

---

## Author Comment (AC2)

Dear Referee #2,

We appreciate your questions and suggestions for improvements. They have been adopted and/or further discussed as described in the following. All responses are given in italic and green font.

Sincerely yours

T. Ning and G. Elgered

P 2, L 36: "...The temporal resolutions of such comparisons are to our knowledge so far limited to 1 h for WVRs (Lu et al., 2016), 2 h for VLBI (Steigenberger et al., 2007), and 6 h for numerical weather models (Zus et al., 2019)." I think at this point it is worth to mention the following recent paper: Kačmařík, M., Douša, J., Zus, F., Václavovic, P., Balidakis, K., Dick, G., and Wickert, J.: Sensitivity of GNSS tropospheric gradients to processing options, Ann. Geophys., 37, 429–446, https://doi.org/10.5194/angeo-37-429-2019, 2019. In this paper hourly nwm/gnss data were utilized in the comparisons.

*Response: The suggested reference has been added in the updated manuscript.*

P2, L55: "...We used the Vienna Mapping Function 1 (VMF1) (Boehm et al.,552006) to map the zenith delay and the gradient mapping function was the one presented by Bar-Sever et al. (1998)" Please provide the formulas for the wet delay gradients somewhere in the manuscript. Maybe this is the right point to do so. For many readers it is not clear what you mean by wet delay gradients. Also the formulas will help to understand why the estimated wet delay gradients depend on the observation geometry, elevation cut off angle, elevation dependent weighting etc.

*Response: The formula has been added in the updated manuscript.*

P3, L65: "...In order to compare to the wet component inferred by the WVR, we subtracted the hydrostatic component computed from the reanalysis product of the European Centrefor MediumRange Weather Forecasts (ECMWF), ERA5, from the total gradient to get the GNSS wet gradient…" Please provide the details (or reference) on how you calculated the hydrostatic gradient from the weather model. What about the wet gradient? Is it worth to be included in your comparisons?I mean is it close to the wvr and the gnss estimates?

*Response: The following text is added describing the calculation of the hydrostatic gradient:*
*"The gradients were calculated from ERA5 by vertical integration of the horizontal refractivity gradients times the height. The profile closest to the site was used together with one profile to the east and one profile to the north to calculate the refractivity gradient profiles."*

*The ERA5 wet gradients have also been compared to the WVR gradients. Using one year of data, the RMS differences are 0.69 mm and 0.66 mm for the east and north gradients, respectively. The corresponding correlation coefficients are 0.51 and 0.52, respectively. Because the WVR-GNSS agreement is better, and that we want a temporal resolution of 5 min, the ERA5 wet gradients are not used further in this study. Since this question came up we add the following sentence after the description of the ERA5 hydrostatic gradient:*
*"We did not use wet gradients from ERA5 because we want to study the gradients with a temporal resolution down to 5 min."*

P4, L95: You make use of a four parameter model to obtain tropospheric parameters. This is very similar to what is done with the gnss. Why do you not apply similar constraints?

*Response: The strength of the WVR data is that gradients can be relatively accurately estimated for a 5 min period independently of the atmospheric variability during adjacent periods. Applying constraints to the WVR data is similar to a low pass filter and will reduce the short term variation (including both rapid atmospheric variability and instrument noise). In this study we strive for a high temporal resolution which would decrease if constraints were to be applied. In the response also to Reviewer #1 we write that the following text will be added at the end of Section 2.2:*

*"The study does not need to assume that the WVR gradients are more accurate compared to the GNSS ones. The main advantage of the WVR gradients is that they are independent and by comparing these to the gradients from different GNSS solutions we can assess the different GNSS processing methods. Furthermore, since we want to study the agreement with as high temporal resolution as possible, we do not apply constraints to the individual 5 min gradients in order to have them independent from adjacent estimates in terms of the atmospheric signals."*

P5: In figure 3 you show the gnss installation. Can you provide a multipath map for the two installations? What do you think is the main limiting factor for the accuracy of the gnss gradient estimates?

*Response: A new figure has been produced (Figure 7) and included in the updated manuscript in order to show the mean code multipath RMS calculated from ONS1 and ONSA.*
*We interpret that the main limiting factor of the GNSS gradient estimates could be an inhomogeneous spatial sampling of the sky which is important when we assume that the linear model describing horizontal gradients has deficiencies. This is especially the case in the north-south direction and that is why in the manuscript we got lower agreement between WVR and GNSS gradients for the north-south direction than the east-west direction.*

P5: In figure 4 you show the observation geometry that is used to estimate the gradients. With such measurements you could investigate the role of the observation geometry, right? For example, you could remove certain observations to the north to get an idea on how well the gradient can be estimated with the limited observation geometry of the gnss.

*Response: Thanks for the suggestions. Such studies need to be carried out systematically in order to get a solid conclusion. Due to the time limit for the revision, we decided not to include such investigation in the current manuscript. Such studies may also be of interest to other scientists. Therefore, we add the following text at the end of the updated manuscript to suggest a future work:*

*"In addition, the role of the geometry of GNSS observations (see Figure 1) can be further studied. For example, one can remove observations in a certain direction and investigate the change of the estimated gradients and their formal uncertainties for different observation geometries."*

Figure 10: Do you have an explanation for the significant bias in the zwd? Out of curiousity: did you compare the measured wvr zwd (measured in zenith direction) with the estimated wvr zwd (estimated by least square adjustment)?

*Response: The bias in the ZWD varies with time and is caused by a combination of algorithm errors and instrumental errors. Because of the gain jumps in the WVR during 2019 the data set is not optimum for studies of the absolute accuracy in the ZWD.(This is one reason why we do not present ZWD comparison results and only focus on the gradients). Over the whole year the mean ZWD difference between ONS1 – WVR is 4.6 mm with a standard deviation (SD) of 6.1 mm. To put these values into a broader context, the corresponding numbers published by Ning et al. (2012) were a ZWD mean difference between ONSA – WVR over 10 years of 0.3 mm and a SD of 6.6 mm.*

*In order to be clear we will add the following sentence in the Section 2.2:*

*"Because of the gain jumps this data set is not optimum for ZWD comparisons on an absolute scale but our focus is on gradients."*

*Ning, T., Haas, R., Elgered, G., and Willén, U.: Multi-technique comparisons of ten years of wet delay estimates on the west coast of Sweden, J. Geod., 86(7), 565--575, https://doi.org/10.1007/s00190-011-0527-2, 2012.*

Figure 11: This is a very interesting plot. again, my question is the following: what would happen if you apply constraints in the wvr estimation procedure.

*Response: We have not carried out such a study (see response above, P5, L95) but we may guess that it will be similar to applying a low-pass filter, something like when we smooth the time series using the Gaussian window to decrease the temporal resolution.*